# Improved Convolutional Pose Machines for Human Pose Estimation Using Image Sensor Data

**DOI:** 10.3390/s19030718

**Published:** 2019-02-10

**Authors:** Baohua Qiang, Shihao Zhang, Yongsong Zhan, Wu Xie, Tian Zhao

**Affiliations:** Guangxi Key Laboratory of Trusted Software, Guilin University of Electronic Technology, Guilin 541004, China; qiangbh@guet.edu.cn (B.Q.); shihao_zhang@yeah.net (S.Z.); xiewu588@126.com (W.X.); zhao_tian3300@163.com (T.Z.)

**Keywords:** human pose estimation, convolutional pose machines, GoogLeNet, fine-tuning, image sensor

## Abstract

In recent years, increasing human data comes from image sensors. In this paper, a novel approach combining convolutional pose machines (CPMs) with GoogLeNet is proposed for human pose estimation using image sensor data. The first stage of the CPMs directly generates a response map of each human skeleton’s key points from images, in which we introduce some layers from the GoogLeNet. On the one hand, the improved model uses deeper network layers and more complex network structures to enhance the ability of low level feature extraction. On the other hand, the improved model applies a fine-tuning strategy, which benefits the estimation accuracy. Moreover, we introduce the inception structure to greatly reduce parameters of the model, which reduces the convergence time significantly. Extensive experiments on several datasets show that the improved model outperforms most mainstream models in accuracy and training time. The prediction efficiency of the improved model is improved by 1.023 times compared with the CPMs. At the same time, the training time of the improved model is reduced 3.414 times. This paper presents a new idea for future research.

## 1. Introduction

Human pose estimation is mainly used to detect the key points of the human body (such as the joints and trunk) from images or videos that come from an image sensor or video sensor. Through human pose estimation, human skeleton information can be described by several key points. For example, given photos of the human body as inputs, a pose estimation model can generate the coordinates of the key points of a human skeleton in these photos. It can be easily seen that human pose estimation plays a very important role in describing human posture and predicting human behavior.

Human pose estimation is not only one of the basic algorithms of computer vision, but is also a fundamental one in many related fields, such as behavior recognition, action recognition [1], character tracking, and gait recognition. Specific applications mainly focus on intelligent video surveillance, patient monitoring systems, human-computer interactions, virtual reality, human animation, smart homes, intelligent security, athlete training, and so on. Human pose estimation algorithms can be divided into three categories: Algorithms based on global features, algorithms based on graphical model, and algorithms based on deep learning [2,3]. After decades of research, human pose estimation methods have achieved good results. However, human pose estimation algorithms based on global features or graphical models [4,5,6] use hand-crafted image features, which mainly rely on the prior knowledge of the designer. Because of the dependence on manual tuning parameters, a lot of manual parameters are very cumbersome to adjust. So, image features only allow a small number of parameters, thus making it difficult to comprehensively capture true human body features. The biggest difference between deep learning (DL) and traditional pattern recognition is that DL automatically learns features from large data rather than using hand-crafted features. So, DL can accurately represent true human body features. At the same time, due to its deeper level, DL has a stronger expression ability [7]. Beginning in 2014, the focus of human pose estimation has shifted to the utilization of DL. In 2015, Pfister proposed Convnet [8], which formulated the human pose estimation as a detection problem and treated the output as a heatmap. However, the model can only detect human skeleton key points of the upper half body and the detection range is limited. In 2016, DeepCut [4] and the later improved DeeperCut [9], were used to detect key points of the human skeleton throughout the body. Both the detection accuracy and speed were improved. Convolutional pose machines (CPMs) [10] have a strong robustness and the detection accuracy is very high on the standard datasets of human pose estimation, such as the Max Planck Institut Informatik (MPII) Human Pose dataset [11] and Leeds Sports Pose (LSP) dataset [12]. However, CPMs have a relatively long training time and low detection speed, which prevents its application in real-time tasks. The Stacked Hourglass [13] of the same period also achieved very good detection results on the standard datasets of human pose estimation. The models of Multi-context [14], Self Adversarial Training [15], and Learning Feature [16] of 2017 and the excellent models, which have further improvements in accuracy, of 2018 are basically new models based on the improved Stacked Hourglass. However, the common drawback of these models is that the models have a great number of parameters, which makes the training time quite long. Besides, the accuracy of the models is still unsatisfactory. Therefore, the following research mainly includes improvements of the network structure to boost the accuracy of human pose estimation, reduce the cost of the model training, and reduce the model parameters. 

The CPMs proposed in 2016 have a strong robustness and the detection accuracy is very high on the standard datasets of human pose estimation. Many subsequent methods are based on this model. In the 2014 ImageNet Large Scale Visual Recognition Challenge (ILSVRC) competition, GoogLeNet [17] won first place. Its success proved that more convolutions and a deeper network can obtain better prediction results. Thanks to its inception structure, GoogLeNet has fewer parameters than other models in the same period. To design a new human pose estimation model with fewer parameters and a higher detection accuracy, in this paper, we choose the CPMs as a base and combine it with GoogLeNet. The first stage of the CPMs is a basic convolutional neural network that directly generates a response map of each human skeleton’s key points from images. We redesign some layers in the GoogLeNet to redesign the first stage of the CPMs. On the one hand, the improved model uses deeper network layers and a more complex network structure to enhance the ability of the first stage to extract low level features, and apply a fine-tuning strategy. On the other hand, the improved model uses the inception structure to greatly reduce the parameters of the model. Thanks to the inception structure characteristics of GoogLeNet, there are fewer parameters and a more complex network structure. Finally, experiments show that our improved model has a higher detection accuracy, fewer model parameters, and a faster detection speed than CPMs and most other mainstream human pose estimation models.

The innovations of this paper are as follows: (1) Our improved model uses deeper network layers and more complex network structures to enhance the ability of low-level feature extraction; and (2) our improved model applies a fine-tuning strategy. 

The rest of this paper is structured as follows. The main idea of the CPMs, and the design, training, and testing of the improved CPMs is illustrated in Section 2. The experiments’ results for two benchmark datasets are presented and discussed in Section 3 and Section 4, respectively. Finally, the conclusions of this work are summarized in Section 5.

## 2. Improved Convolutional Pose Machines

This chapter is divided into two parts. In the first part, we provide a brief introduction to the main idea of convolution pose machines. In the second part, we describe the details of the design, training, and testing of the improved CPMs.

### 2.1. Convolutional Pose Machines

In this section, we provide a brief introduction to the major idea of convolution pose machines.

#### 2.1.1. Pose Machines

We denote the pixel location of the $q^{-\mathrm{th}}$ (the q is between 0 and 14 in this paper) anatomical landmark (which we refer to as a part), $Y_{q}\in U\subset R^{2}$, where U is the set of all (x,y) locations of an image. We aim to predict the image locations, $Y=\left(Y_{1},\;Y_{2},\;\ldots ,\;Y_{Q}\right)$, for all Q parts. A pose machine [18] (see Figure 1a,b) consists of a sequence of multi-class predictors, $g_{s}\left(\cdot \right),$ that are trained to predict the location of each part in each level of the hierarchy. In each stage s∈{1,…,S}, the classifiers, $g_{s}$, predict the beliefs for assigning a location to each part, $Y_{q}=u,\forall u\in U$, based on features extracted from the patch of the location, u, denoted by $v_{u}\in R^{c}$ and contextual information from the preceding classifier in the neighborhood around each in stage s. A classifier produces the following belief values in the stage, *s* = 1:(1)$$g_{1}\left(v_{u}\right)\to \left\{d_{1}^{q}\left(Y_{q}=u\right)\right\}{\;}_{q\in \left\{0,\ldots ,Q\right\}},$$where $d_{1}^{q}\left(Y_{q}=u\right)$ is the score predicted by the classifier, $g_{1}$, for assigning the $q^{th}$ part in the stage, *s* = 1, at the image location, u. We represent all the beliefs of part q evaluated at every location, $u={\left(x,y\right)}^{S}$, in the image as $d_{s}^{q}\in R^{w\times h}$, where w and h are the width and height of the image, respectively. That is:(2)$$d_{s\;}^{q}\left[x,y\right]=d_{s}^{q}\left(Y_{q}=u\right).$$

In the follow-up stages, the classifier, $g_{s}$, predicts a belief for assigning a location to each part, $Y_{q}=u,\forall u\in U$, based on (1) features of the image data, $v_{z}^{s}\in R^{c}$, again, and (2) contextual information from the preceding classifier, $g_{s-1}$, in the neighborhood around each $Y_{q}$:(3)$$g_{s}\left(v_{u}^{\prime },F_{s}\left(u,d_{s-1}\right)\right)\to \left\{d_{s}^{q}\left(Y_{q}=u\right)\right\}{\;}_{q\in \left\{0\ldots Q+1\right\}},$$where $F_{s>1}\left(\cdot \right)$ is a mapping from the beliefs, $d_{s-1}$, to context features. In each stage, the computed beliefs provide an increasingly accurate estimate for the location of each part. Note that we permit image features, $v_{u}^{\prime }$, for the follow-up stage to be different from the image feature, v, used in the stage, *s* = 1. The pose machine proposed in [10] used boosted random forests for prediction ($\left\{g_{s}\right\}$), fixed hand-crafted image features across all stages ($v^{\prime }=v$), and fixed hand-crafted context feature maps ($F_{s}\left(\cdot \right)$) to capture the spatial context across all stages.

#### 2.1.2. Convolutional Pose Machines

The CPM is a convolutional neural network for human pose estimation on single 2D human pose estimation datasets, such as MPII, LSP, and Frames Labeled In Cinema (FLIC). The model uses CNN [19,20,21] for human pose estimation. Its main contribution lies in the use of sequential convolution architecture to express spatial information and texture information [10]. The sequential convolution architecture can be divided into several stages in the network. Each stage has a part of the supervised training [17,22], which avoids the problem of gradient disappearance in the deep network [23,24,25,26]. In the first stage, the original image is used as input. In the later stages, the feature map of the first stage is used as input. The main purpose is to fuse spatial information, texture information, and central constraints. In addition, the use of multiple scales to process the input feature map and response map for the same convolution architecture not only ensures accuracy, but also considers the distance relationship between the key points of each human skeleton.

The overall structure of the CPMs is shown in Figure 2. In Figure 2, the “C”, “MC1, MC2, …” means different convolution layers, and the “P” means different pooling layers. The “Center map” is the center point of the human body picture, and it is used to aggregate the response maps to the image centers. The “Loss” is the minimum output cost function, and it is the same as the “Loss” of the subsequent figures.

The first stage of the CPMs is a basic convolutional neural network (white convs) that directly generates the response map of each human skeleton’s key points from images. The whole model has the response maps of 14 human skeleton key points and a background response map, with a total of 15 layers of response maps.

The network structure with the stage ≥2 is completely consistent. A feature image with a depth of 128, which is from stage 1, is taken as the input and three types of data (texture features, spatial features, and center constraints (the center point of the human body picture is used to aggregate the response maps to the image centers)) are fused by the concat layer.

The original color image and some feature maps with a depth of 128 (the overlay of 128 heatmaps) are shown in Figure 3 below.

### 2.2. Improved Convolutional Pose Machines

#### 2.2.1. Design of the Improved Convolutional Pose Machines 

There are two types of CPMs models designed by Shih-En Wei. One is the original CPM-Stage6 model and the other is the VGG10-CPM-Stage6 model based on the VGGNet-19 design (they are both models that the authors publicly exposed). In the author’s publicly trained models, the VGG10-CPM-Stage6 model has a faster model training speed, fewer model parameters, and higher accuracy on the same verification dataset than the CPM-Stage6 model. Even so, the feature extractor of the VGG10-CPM-Stage6 model used for fine-tuning is still large and after combining multiple large nuclear layers of multiple stages, the computational complexity of the model becomes very significant both in deployment and training. It is more difficult. The VGG10-CPM-Stage6 model has many parameters and its convergence speed is not fast enough. Besides, its network layer is not deep enough, so its learning ability is not strong enough. To improve the detection accuracy of the model and speed up the convergence of the model, an effective way is to increase the depth of the network and the number of convolution layers, while reducing the parameters of the network model.

In the 2014 ILSVRC competition, GoogLeNet achieved first place. Its success proved that more convolutions and a deeper network can obtain better prediction results. Because of its inception structure, the GoogLeNet model has fewer parameters than other models in the same period. To design a new human pose estimation model with fewer parameters and a higher detection accuracy, we attempted to combine CPMs with GoogLeNet. In this paper, we redesigned some layers of GoogLeNet to redesign stage 1 of CPMs. Specifically, we selected different inception layers, Inc(4a) (the first nine layers of GoogLeNet), Inc(4b) (the first 11 layers of GoogLeNet), Inc(4c) (the first 13 layers of GoogLeNet), Inc(4d) (the first 15 layers of GoogLeNet), and Inc(4e) (the first 17 layers of GoogLeNet), of GoogLeNet separately for stage 1 of the new human pose estimation models. So, there are five new models. The overall structure of these improved models is shown in Figure 4. In Figure 4, the “C”, “4-3C, 4-4C…”, “MC1, MC2, …” means different convolution layers, and the “P” means different pooling layers. The “Center map” is the center point of the human body picture, and it is used to aggregate the response maps to the image centers. Most of the new models increase the number of convolution layers and use a more complex network structure to enhance the ability of stage 1 to extract low level features of images. At the same time, they apply a fine-tuning strategy. Thus, they can further improve the accuracy of detection. Besides, the new models use the inception structure to greatly reduce the parameters of model. Thus, the convergence speed of the model training is also significantly improved. These models use fine-tuning training on multiple real human pose estimation datasets and then the Extended LSP verification dataset is selected for verification. Finally, the experiments show that these new models can not only further improve the accuracy of detection, but also greatly reduce the amount of parameters, and shorten the model training time.

The main data changes of the stage 1 network of the improved models are shown in Table 1.

The main data changes of the stage ≥2 network of the new models are shown in Table 2.

Table 1 and Table 2 show the changes of the feature maps with the depth of the network. Specifically, the structure of the different stages (stage ≥ 2) are exactly the same, except that the content of the concat_Stage fusion has changed locally.

The verification accuracy comparisons (verification on the LSP dataset; we used PCK@0.2 for the evaluation of the GoogLeNet13-CPM-Stage6) between the improved models based on different inception layers and Shih-En Wei’s VGG10-CPM-Stage6 model are shown in Figure 5 below.

The parameters’ quantity (verification on the LSP dataset: The CPM-Stage6 and VGG10-CPM-Stage6 have many parameters; to display them better in Figure 6, they were divided by 10) comparisons between the improved models based on the different inception layers and Shih-En Wei’s VGG10-CPM-Stage6 model are shown in Figure 6 below.

We observed that the parameters of the new human pose estimation models designed by different layers of GoogLeNet are much fewer than the parameters of Shih-En Wei’s VGG10-CPM-Stage6 model. Thus, it requires less training time and less training cost at the training stage, as well as the verification accuracy being promoted. The above experiments also validate the influence of different Inception layers on the detection effect of the designed human pose estimation model. We found that with the increase of the number of layers of inception, the accuracy of the human pose estimation model showed a trend of slowly rising to a certain extent and then slowly declining after reaching the peak. The accuracy was maintained above 0.8770, with the highest accuracy of 0.8841. After analysis of the reasons, the GoogLeNet was originally a successful model for image classification [27] and our improved model is a model for human pose estimation. With the increasing number of inception layers used in the new human pose estimation models, the new models enhance the ability to extract low level features on human pose estimation datasets to improve the detection effect. However, as the number of inception layers in the GoogLeNet continues to increase, the simple and low level features that are learning in the GoogLeNet slowly transform into the learning of deep complex and specific image classification features. It is very different from the features of deep complex and specific human pose estimation. Thus, this is influential. In the experiments, the previous layers of GoogLeNet, Inc(4c) (the first 13 layers of GoogLeNet), were chosen to redesign the GoogLeNet13-CPM-Stage6 with the highest accuracy. Therefore, this model is also the best model to be used in the following experimental chapters.

#### 2.2.2. Training and Testing

In the actual training process, if the models are trained from the beginning, the problem of gradient dispersion easily occurs. Therefore, this paper uses the method of fine-tuning [28,29] to train the models on the MPII Human Pose training dataset or the Extended LSP training dataset, and then the trained models were used to perform verification tests on the Extended LSP verification dataset. The whole process is mainly divided into the following steps:

(1) After construction of the improved models, the GoogLeNet’s parameters are used, which is trained in the ImageNet competition to initialize the parameters of the previous layers of the improved models;

(2) the MPII training dataset or the Extended LSP training dataset is input into the improved models according to the batch_size, and “stepsize=120,000” is used to adjust the learning rate. As the models continue to train, the learning rate is adjusted every 120,000 times;

(3) during the training process, the loss value of each stage of the new models is continuously reduced until it is stable. The verification or test requires a separate code, so the accuracy of the models’ detection is not verified during training;

(4) the loss value of each stage and the total loss value of the improved models is output, and the models are saved once every 5000 iterations; and

(5) each trained model is selected and is verified on the Extended LSP verification dataset, and the accuracy of the 1000 image verification and other verification index information records is completed. The model with the best verification indicators is selected.

The setting of the main parameters during the training of the improved models is shown in Table 3.

Compared with the Shih-En Wei optimal state VGG10-CPM-Stage6, the improved models can increase the depth of the network and greatly reduce the parameter quantity of the previous layers of the network. Thereby, the expression ability of the improved models can be enhanced and the time of the models’ training can be effectively shortened.

### 2.3. Learning in GoogLeNet13-CPM-Stage6

Deep neural networks that are training tend to produce gradient disappearance. As mentioned in Bradley [24] and Bengio et al. [25], the intensity of the gradient decline in backpropagation is affected by the number of intermediate layers between the input and output layers.

Fortunately, the sequence prediction framework of GoogLeNet13-CPM-Stage6 naturally trains deep models and each stage continuously generates a response map of the key points of each human skeleton. We define a loss function, f, at the output of each stage,s, to minimize the $l_{2}$ distance between the predictive response maps of each key points of the human skeleton and its true annotated response maps, thus guiding the network model to achieve a desired effect. The true annotated response map for a part, q, is recorded as $d_{*}^{q}\left(Y_{q}=u\right)$. The true annotated response map can be constructed by placing a Gaussian peak at the true coordinate position of each human skeleton key point, q. We define the minimum output cost function of each stage as:(4)$$f_{s}=\sum_{q=1}^{Q+1}\sum_{u\in U}||d_{s}^{q}\left(u\right)-d_{*}^{q}\left(u\right)||{\;}_{2}^{2}.$$

The overall objective for the full architecture is obtained by setting the losses at each stage and is given by:(5)$$F=\sum_{s=1}^{S}f_{s}.$$

In the actual training process, we use the standard stochastic gradient descent method to train all the S stages in the network. To share the image feature, v’, in all follow-up stages, we share the weights of the corresponding convolutional layers (see Figure 2) in the stages, s ≥ 2.

## 3. Experimental Results

### 3.1. Experimental Environment and Datasets

In our experiments, we used an Intel Xeon E5-2698 V4 (20 cores) processor with a 50 GB memory. We used a single NIVDIA Tesla P100 graphics card. We selected the 64-bits Ubuntu 14.04 operating system, Caffe deep learning framework, and Python 2.7 as the development environment. We also utilized the following tools: PyCharm 2017.1.2.

In this paper, we used three benchmark datasets for human pose estimation, the MPII Human Pose dataset [11], Extended LSP dataset [30], and LSP dataset [12], which came from an image sensor and labelled well. 

The MPII Human Pose dataset includes around 25,000 images containing over 40,000 people with annotated body joints. The images were systematically collected using an established taxonomy of every day human activities. Overall, the dataset covers 410 human activities and each image is provided with an activity label. The MPII Human Pose dataset is divided into 25,000 training human samples and 3000 validated human samples. Each sample contains the identification (ID) of the sample image, the coordinate information of the center points of the sample, the true coordinate information of the key points of the human skeleton, and so on.

The Extended LSP dataset contains 10,000 images gathered from Flickr searches for the tags, ’parkour’, ’gymnastics’, and ’athletics’, and consists of poses deemed to be challenging to estimate. Each image has a corresponding annotation gathered from Amazon Mechanical Turk and as such cannot be guaranteed to be highly accurate. Each image was annotated with up to 14 visible joint locations. The LSP dataset contains 2000 pose annotated images of mostly sports people gathered from Flickr using the tags shown above. The Extended LSP dataset and the LSP dataset were divided into 11,000 training human samples and 1000 validated human samples. Each sample also contains the ID of the sample image, the true coordinate information of the key points of human skeleton, and so on. 

The basic information of the datasets is shown in Table 4.

### 3.2. Experimental Procedure

To validate the generalization capabilities [31] and prediction accuracy of our improved model, we designed three sets of comparative experiments.

In the first set of experiments, we trained our GoogLeNet13-CPM-Stage6 on the MPII Human Pose training dataset and then validated it on the Extended LSP validation dataset. In the contrast experiment, two models trained by Shih-En Wei on the MPII Human Pose training dataset, CPM-Stage6 and VGG10-CPM-Stage6, were selected.

In the second set of experiments, we trained our GoogLeNet13-CPM-Stage6 on the Extended LSP training dataset and then validated it on the Extended LSP validation dataset. In the contrast experiment, most leading models of human pose estimation on the Extended LSP verification dataset were selected.

In the third set of experiments, we trained our GoogLeNet13-CPM-Stage6 on the MPII Human Pose training dataset and then validated it on the MPII Human Pose validation dataset. In the contrast experiment, most leading models of human pose estimation on the MPII Human Pose dataset were selected.

To evaluate these models, we used the proportion of correctly predicted key points (PCK) as a metric on the validation dataset.

Generally speaking, when the distance between the predicted coordinates of the key points of a human skeleton and the true coordinates of the key points of a human skeleton is less than a certain proportion (a) of the pixel length of the human head or trunk in the image, it is considered to be correct. This evaluation method is called PCK@a. 

According to PCK@a, the total number of key points of a human skeleton that are predicted to be correct is recorded as TP and the total number of key points of a human skeleton that are predicted to be incorrect is recorded as FN, so the calculation formula of the verification accuracy is as shown in (6):(6)$$accuracy_{PCK@a}=\frac{TP}{TP+FN}.$$

For the Extended LSP validation dataset, the validation accuracy on PCK@0.2 is the main criterion for the evaluation of the GoogLeNet13-CPM-Stage6, and for the MPII Human Pose validation dataset, the validation accuracy on PCKh@0.5 is the main criterion for the evaluation of the GoogLeNet13-CPM-Stage6.

### 3.3. Experimental Results

For the first set of experiments in Section 3.2, the accuracy of the three models on the Extended LSP verification dataset is shown in Table 5.

From Table 5, we observed that our improved model, which trained 175,000 times on the MPII Human Pose training dataset, would obtain an ideal model. It showed the fastest convergence speed and the highest accuracy compared to those models of Shih-En Wei’s open trained one. 

The speed of convergence, training time, and average detection time of the three models were compared as shown in Table 6.

From Table 6, we observed that our improved model has the fastest convergence speed based on the time it takes to complete the model training, the least training time, and the fastest detection speed compared to those models of Shih-En Wei’s open trained one, because our improved model increases the depth of the network of stage 1 and uses a more complex network structure to extract low level features of images. Meanwhile, it applies fine-tuning strategy. Thus, it obtains a higher accuracy of detection and enhances the generalization ability of the model. Besides, the improved model uses the inception structure to greatly reduce the parameters of model. Thus, the convergence speed of the model training was also significantly improved. At the same time, it greatly shortens the training time, and reduces the average detection time of a single image.

For the second set of experiments in Section 3.2, the accuracy of the nine models on the Extended LSP verification dataset is shown in Table 7. Although our improved model could detect 14 key points (head, neck, right shoulder, left shoulder, right elbow, left elbow, right wrist, left wrist, right hip, left hip, right knee, left knee, right ankle, left ankle; the 14 key points are shown in Figure 7 below.), in Table 7, we adopt a unified approach with seven key points (the results (comparisons) in the homepage of the MPII Human Pose Dataset) to compare most mainstream models more conveniently. This shows the average detection results of the left and right key points (e.g., left knee, right knee).

From Table 7, we observed that our improved model* achieves a high level of accuracy compared with most other leading human pose estimation models. Compared with Wei et al, the overall PCK increased by 2.1%. In the head, shoulder, elbow, and wrist, the accuracy of our improved model* is also at a high level compared with most other leading models. In the hip, knee, and ankle, there are slight gaps for the accuracy of our improved model* compared with some other leading models. Overall, it is one of the leading models.

The speed of convergence, training time, and the average detection time of the six models are shown in Table 8.

From Table 8, we observed that our improved model* has the fastest convergence speed, the least training time, and the fastest detection speed compared with most other leading models. The reasons are 1) we introduced the inception structure to our improved model to greatly reduce parameters; and 2) our model applied a fine-tuning strategy. Thus, it is easier to train and detection.

For the third set of experiments in Section 3.2, the accuracy of the nine models on the MPII Human Pose estimation verification dataset is shown in Table 9.

From Table 9, we observed that our improved model achieves a high level of accuracy compared with most other leading human pose estimation models. Compared with Wei et al, the overall PCK increased by 3%. In the head, shoulder, and wrist, the accuracy of our improved model is also the highest compared with most other leading models. In the hip, knee, and ankle, there are slight gaps for the accuracy of our improved model compared with some other leading models. Overall, it is much better than most other leading models.

## 4. Discussion

Extensive experimental results show that to improve the network structure of the model to obtain a higher detection accuracy, reduce the parameters of the model, and reduce the cost of model training, a new network model based on the combination of a high accuracy of image classification model, GoogLeNet, with an excellent human pose estimation model, must be designed.

Our improved convolutional pose machines can be applied to the following areas, such as behavior recognition, character tracking, gait recognition, etc. Specifically, it mainly focuses on intelligent video surveillance, patient monitoring systems, human-computer interaction, virtual reality, human animation, smart homes, intelligent security, athlete training, and so on. Although our improved model could obtain a high accuracy and a very fast detection speed, it is still not in real-time. Because real-time performance is required for human pose estimation in the field of videos, our improved model is more suitable for images that are sourced from an image sensor. 

Regarding novelty, we also combined the CPMs with the Resnet [35] to redesign some new models. Unfortunately, although the depth of the Resnet is deeper than GoogLeNet, the detection results of these new models are not ideal. At the same time, the parameters’ quantity of these new models is also larger. Besides, the Inception v2 [36] and Inception v3 [37] were also considered by us. Because the structure of GoogLeNet (Inception v1) is very different from them, we studied the structure of them carefully and found that it is impossible to combine CPMs with them directly. Therefore, in the future, we will mainly conduct the following work: (1) We will continue to try to reduce the parameters of the model to improve the detection speed of the model; (2) the CPMs and Stacked Hourglass are both popular methods in 2016 and we will introduce the inception modules in the Stacked Hourglass for further research.

## 5. Conclusions

Our GoogLeNet13-CPM-Stage6 innovatively combines the classic GoogLeNet model, which has a high accuracy of image classification, with the CPMs model, which is an excellent human pose estimation model. Compared with the two models of Shih-En Wei and most other mainstream human pose estimation models, the GoogLeNet13-CPM-Stage6 obtained a higher detection rate and shortened the average detection time of a single image. Meanwhile, the training time of the model was also reduced. Our improved model is the same as most mainstream human pose estimation models, which are independent from the user. Extensive experiments on several datasets show that our improved model has a very high detection accuracy. Besides, it also achieved perfect results in more complex scenes.

Human pose estimation is still an active research component of the field of computer vision. Existing algorithms of human pose estimation have not achieved perfect results and there are still some incorrect detection cases in more complex scenes. Through experiments, we identified that the combination of a model, with a high image classification accuracy or good image detection effect, with an excellent human pose estimation model to design a new network and apply a fine-tuning strategy will be more effective for human pose estimation. This conclusion provides some guidance for future research.

## Figures and Tables

**Figure 1 sensors-19-00718-f001:**
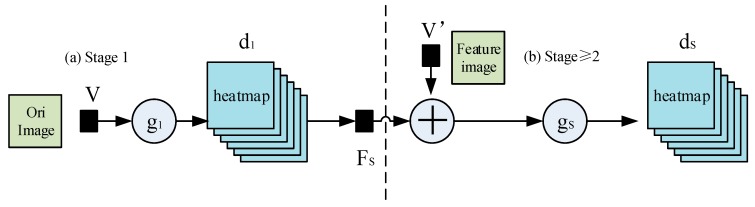
Pose machines.

**Figure 2 sensors-19-00718-f002:**
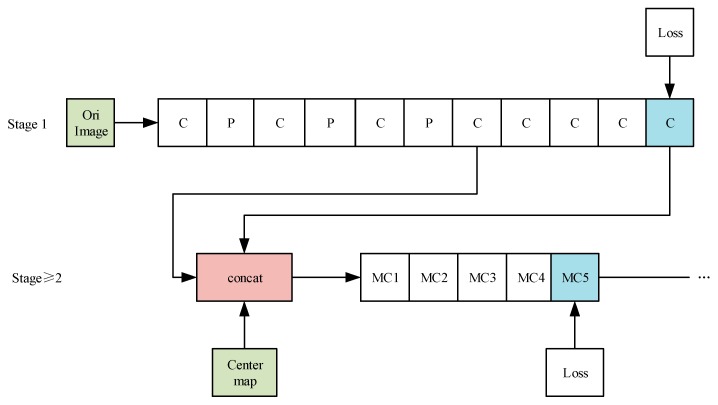
The overall structure of the convolutional pose machines.

**Figure 3 sensors-19-00718-f003:**
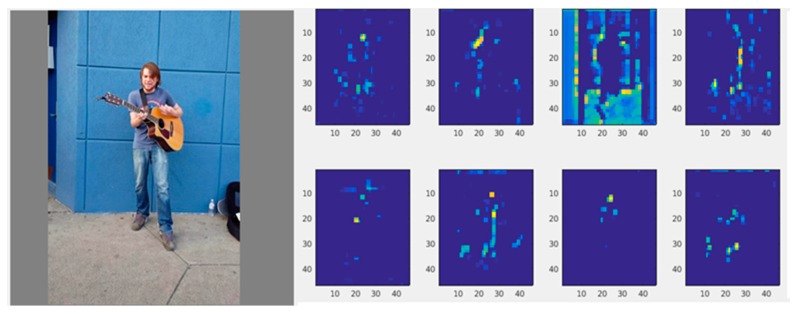
The original color image and some feature maps with a depth of 128.

**Figure 4 sensors-19-00718-f004:**
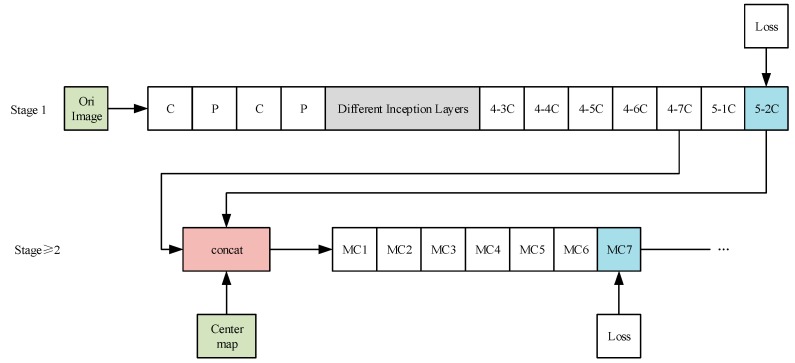
The overall structure of the improved models.

**Figure 5 sensors-19-00718-f005:**
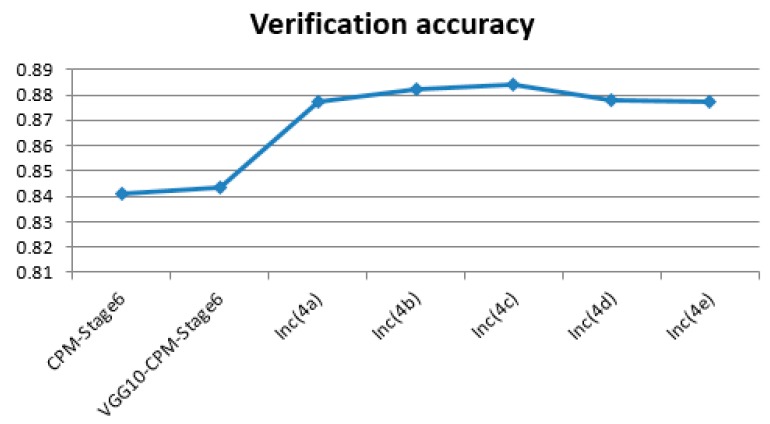
The verification accuracy comparisons of these models.

**Figure 6 sensors-19-00718-f006:**
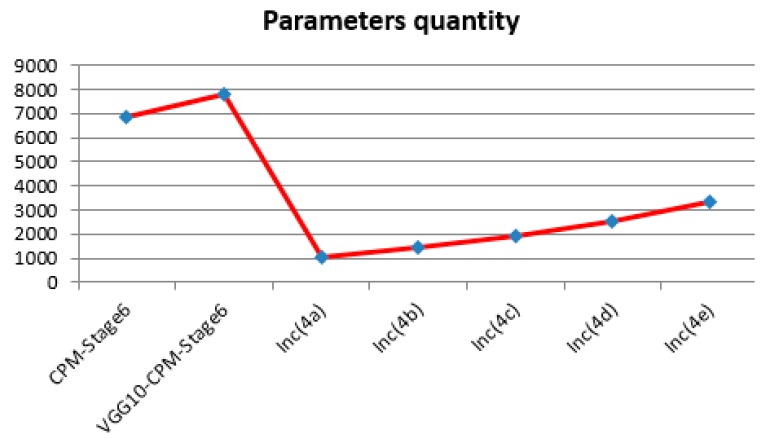
The parameters’ quantity comparisons of these models.

**Figure 7 sensors-19-00718-f007:**
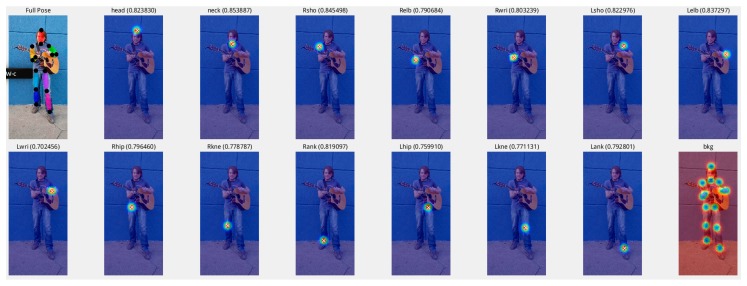
The 14 key points.

**Table 1 sensors-19-00718-t001:** The main data change of the stage 1 network.

Type	Kernel Size/Stride Size	Depth	Output Size
conv1	7 × 7/2	1	184 × 184 × 64
max pool	3 × 3/2	0	92 × 92 × 64
conv2	3 × 3/1	2	92 × 92 × 192
max pool	3 × 3/2	0	46 × 46 × 192
Inception(3a)	-	2	46 × 46 × 292
Inception(3b)	-	2	46 × 46 × 480
Inception(4a)	-	2	46 × 46 × 512
Inception(4b)	-	2	46 × 46 × 512
Inception(4c)	-	2	46 × 46 × 512
Inception(4d)	-	2	46 × 46 × 528
Inception(4e)	-	2	46 × 46 × 832
conv4_3_CPM	3 × 3/1	1	46 × 46 × 256
conv4_4_CPM	3 × 3/1	1	46 × 46 × 256
conv4_5_CPM	3 × 3/1	1	46 × 46 × 256
conv4_6_CPM	3 × 3/1	1	46 × 46 × 256
conv4_7_CPM	3 × 3/1	1	46 × 46 × 128
conv5_1_CPM	1 × 1/1	1	46 × 46 × 512
conv5_2_CPM	1 × 1/1	1	46 × 46 × 15

**Table 2 sensors-19-00718-t002:** The main data change of the stage ≥2 network.

Type	Kernel Size/Stride	Depth	Output Size
conv4_7_CPM	3 × 3/1	1	46 × 46 × 128
pool_center_lower	9 × 9/8	0	46 × 46 × 1
conv5_2_CPM	1 × 1/1	1	46 × 46 × 15
concat_Stage2	-	0	46 × 46 × 144
Mconv1_Stage2	7 × 7/1	1	46 × 46 × 128
Mconv2_Stage2	7 × 7/1	1	46 × 46 × 128
Mconv3_Stage2	7 × 7/1	1	46 × 46 × 128
Mconv4_Stage2	7 × 7/1	1	46 × 46 × 128
Mconv5_Stage2	7 × 7/1	1	46 × 46 × 128
Mconv6_Stage2	1 × 1/1	1	46 × 46 × 128
Mconv7_Stage2	1 × 1/1	1	46 × 46 × 15

**Table 3 sensors-19-00718-t003:** The setting of the main parameters.

Parameters	Meaning
batch_size = 16	The size of the training data for a single iteration
Backend = LMDB	Database format
lr_policy = “step” stepsize = 120,000	Learning strategy is step The times of iterations required to adjust the learning rate
weight_decay = 0.0005	Weight attenuation coefficient
base_lr = 0.000080	Initial value of learning rate
momentum = 0.9 max_iter = 350,000	Momentum Maximum iterations

**Table 4 sensors-19-00718-t004:** The basic information of the datasets.

Dataset Name	Category	Num of Key Points	Training/Validation
MPII	Whole body	14	25,000/3000
Extended-LSP + LSP	Whole body	14	11,000/1000

**Table 5 sensors-19-00718-t005:** The accuracy of the three models.

Models	Iteration	Accuracy (1000 Images)
CPM-Stage6	630,000	0.8554
VGG10-CPM-Stage6	320,000	0.8798
Improved Model	175,000	0.8823

**Table 6 sensors-19-00718-t006:** The contrasted details of the three models.

Models	Iteration	Cost Time	Speed (ms/each Image)
CPM-Stage6	630,000	180 h	260.7
VGG10-CPM-Stage6	320,000	91 h	255.6
Improved Model	175,000	36.37 h	181.2

**Table 7 sensors-19-00718-t007:** The accuracy of the nine models.

Models	Head	Shoulder	Elbow	Wrist	Hip	Knee	Ankle	PCK
Lifshitz et al. [32] **	96.8	89.0	82.7	79.1	90.9	86.0	82.5	86.7
Pishchulin et al. [4] *	97.0	91.0	83.8	78.1	91.0	86.7	82.0	87.1
Insafutdinov et al. [9] *	97.4	92.7	87.5	84.4	91.5	89.9	87.2	90.1
Wei et al. [10] *	97.8	92.5	87.0	83.9	91.5	90.8	89.9	90.5
CU-Net-8 [33]	97.1	94.7	91.6	89.0	93.7	94.2	93.7	93.4
Tang et al. [34]	97.5	95.0	92.5	90.1	93.7	95.2	94.2	94.0
Chu et al. [14] *	98.1	93.7	89.3	86.9	93.4	94.0	92.5	92.6
Improved model	96.8	90.5	85.3	81.7	90.3	87.8	86.3	88.4
Improved model *	98.2	93.7	89.8	87.3	92.7	93.8	92.3	92.6

* models trained when adding MPII training set to the LSP training and LSP extended training set. ** models trained when adding MPII training set to the LSP training set.

**Table 8 sensors-19-00718-t008:** The contrasted details of the six models.

Models	Iteration	Cost Time	Speed (ms/each)
Lifshitz et al. [32] **	-	-	700
Pishchulin et al. [4] *	1,000,000	-	57,995
Insafutdinov et al. [9] *	1,000,000	120 h	230
Wei et al. [10] *	985,000	280 h	260.7
Chu et al. [14] *	-	-	-
Improved model *	275,000	82 h	180.9

* models trained when adding MPII training set to the LSP training and LSP extended training set. ** models trained when adding MPII training set to the LSP training set.

**Table 9 sensors-19-00718-t009:** The accuracy of the nine models.

Models	Head	Shoulder	Elbow	Wrist	Hip	Knee	Ankle	PCKh
Lifshitz et al. [32]	97.8	93.3	85.7	80.4	85.3	76.6	70.2	85.0
Pishchulin et al. [4] *	94.1	90.2	83.4	77.3	82.6	75.7	68.6	82.4
Insafutdinov et al. [9]	96.8	95.2	89.3	84.4	88.4	83.4	78.0	88.5
Wei et al. [10] *	97.8	95.0	88.7	84.0	88.4	82.8	79.4	88.5
Newell et al. [13]	98.2	96.3	91.2	87.1	90.1	87.4	83.6	90.9
Chu et al. [14]	98.5	96.3	91.9	88.1	90.6	88.0	85.0	91.5
CU-Net-8 [33]	97.4	96.2	91.8	87.3	90.0	87.0	83.3	90.8
Tang et al. [34]	97.4	96.4	92.1	87.7	90.2	87.7	84.3	91.2
Improved model	98.6	96.4	91.9	88.5	90.4	87.8	84.8	91.5

* models trained when adding MPII training set to the LSP training and LSP extended training set.

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
