# Peer review of "Improved Convolutional Pose Machines for Human Pose Estimation Using Image Sensor Data"

_sensors, 2019, doi:10.3390/s19030718_

Round 1

Reviewer 1 Report

Summary

The paper uses the inception modules in the convolutional pose machine. Experiments show the accuracy is improved and the parameter number is decreased. The training and inference time is also reduced.

The overall quality of the submission is fine. The authors are suggested to address following questions before considering acceptance.

1. The used inception modules are not clear. What do the symbols like “Inc(3a)~Inc(4d)” mean? Some figures are better given.

2. The figure layout needs some improvements. For example, Figures 5 and 6 contain limited information but use too much space.

3. Accuracy on the MPII dataset is not reported and compared with state-of-the-art methods.

4. Many recent methods are not compared in Table 7.

5. The inception modules are only used in the CPM. How about using it in other methods? Stacked hourglasses are more popular than the CPM currently.

6. Human pose estimation is an extremely active research area. Discussions on more recent work are highly suggested to keep the track of related work:

    [1] Jointly optimize data augmentation and network training: adversarial data augmentation in human pose estimation. CVPR 2018

    [2] Quantized densely connected u-nets for efficient landmark localization. ECCV 2018

    [3] CU-Net: Coupled U-Nets. BMVC 2018 

Author Response

I am very grateful to receive your suggestions. These suggestions are very valuable and very helpful for the improvement of our manuscripts.We have revised the manuscript according to the your suggestion. Here are responses for the suggestion.

Suggestion 1. The used inception modules are not clear. What do the symbols like “Inc(3a)~Inc(4d)” mean? Some figures are better given.

Our response: Because of the inaccuracy of my previous language expression, the “Inc(3a)~Inc(4d)” is replaced by ”Inc(4d)” (The first 15 layers of GoogLeNet). (rows 171-174. Specifically, we have selected different Inception layers, Inc(4a) (The first 9 layers of GoogLeNet), Inc(4b) (The first 11 layers of GoogLeNet), Inc(4c) (The first 13 layers of GoogLeNet), Inc(4d) (The first 15 layers of GoogLeNet), Inc(4e) (The first 17 layers of GoogLeNet), of GoogLeNet separately for stage 1 of new human pose estimation models.) (rows 226-227.)

Suggestion 2. The figure layout needs some improvements. For example, Figures 5 and 6 contain limited information but use too much space.

Our response: The figure 5 and 6 have been revised according to your suggestion now. (rows 203,209)

Suggestion3. Accuracy on the MPII dataset is not reported and compared with state-of-the-art methods.

Our response: We add new experiment that we trained our GoogLeNet13-CPM-Stage6 on the MPII Human Pose training dataset and then validated it on the MPII Human Pose validation dataset now. In the contrast experiment, most leading models of human pose estimation on the MPII Human Pose dataset were selected. (rows 310-313,382).

Suggestion 4. Many recent methods are not compared in Table 7.

Our response: We add some recent methods to compare with our improved model in Table 7 now. (rows 361-362)

Suggestion 5. The inception modules are only used in the CPM. How about using it in other methods? Stacked hourglasses are more popular than the CPM currently.

Our response: Combining a model which is with high image classification accuracy or good image detection effect with CPMs to design a new network is our starting point. We combine the CPMs with the Resnet to redesign some new models. Unfortunately, although the depth of the Resnet is deeper than GoogLeNet, the detection results of these new model are not ideal. At the same time, the parameters quantity of these new model is also larger. Besides, the Inception v2, Inception v3 are also considered by us. Because the structure of GoogLeNet (Inception v1) is very different from them, we studied the structure of them carefully and find that it is not available to combine CPMs with them well. (rows 402-407)

Suggestion 6. Human pose estimation is an extremely active research area. Discussions on more recent work are highly suggested to keep the track of related work:

[1] Jointly optimize data augmentation and network training: adversarial data augmentation in human pose estimation. CVPR 2018

[2] Quantized densely connected u-nets for efficient landmark localization. ECCV 2018

[3] CU-Net: Coupled U-Nets. BMVC 2018

Our response: We add some more recent work [2][3]in Table 7 and Table 9 now. In the future, we mainly do the following work. (1) We will continue to try to reduce the parameters of the model to improve the detection speed of the model. (2) The CPMs and Stacked Hourglass are both popular methods in 2016, we will introduce the inception modules in the Stacked Hourglass for further research.(rows 407-411)

Reviewer 2 Report

My comments and questions are given in the attached file.

Author Response

I am very grateful to receive your suggestions. These suggestions are very valuable and very helpful for the improvement of our manuscripts.We have revised the manuscript according to the your suggestions. Here are responses for the suggestions.

Suggestion 1. According to the title and the discussion (rows 357-358), the proposed method is more suitable for images from image sensor than video. Could you explain why? What is the difference between an image taken from some image sensor and an image from a video?

Our response: Although our improved model could get high accuracy and very fast detection speed, it is still not real-time. Because real-time is required for human pose estimation in the field of video, our improved model is more suitable for images which are from image sensor. Because of the inaccuracy of my previous language expression, we have deleted these words(than video) for accuracy expression now. (rows 398-401)

Suggestion 2. I do not feel qualified to judge about the English language but I think that the sentence starting with "In order to design..." (rows 164-165) is not finished. I have also some doubts about the correctness of some sentences from the Discussion and Conclusions sections (rows 348-351). Therefore I would recommend checking the English language and style in these sections.

Our response: The sentence starting with "In order to design..." have been finished now. (rows 168-170). We also have checked the English language and style in other sections. (rows 391-394)

Suggestion 3. Could you write how many different users performed the poses contained in the two considered datasets? Is it possible to add this information to the datasets description in the rows 272 - 283?

Our response: In this paper, we will use three benchmark datasets for human pose estimation, MPII Human Pose dataset, Extended LSP dataset and LSP dataset which are coming from image sensor and labelled well. These datasets don’t contained the information “how many different user performed the poses?” So we have added the introduction information of these datasets as much as possible.(rows 280-295)

Suggestion 4. In table 7 some abbreviations are used as the columns headers. I guess that they mean the key points of human skeleton. Could you explain how many key points are used? Maybe it is possible to include some drawing? How about left/right key points, eg. Left knee, right knee? They are not distinguished in this table.

Our response: Although our improved model could detect 14 key points (Head, neck, right shoulder, left shoulder, right elbow, left elbow, right wrist, left wrist, right hip, left hip, right knee, left knee, right ankle, left ankle), the 14 key points are showed in figure 7 below, in table 7, we adopt a unified approach (The results (comparisons) in the homepage of the MPII Human Pose Dataset) to compare with most mainstream models conveniently. It will show the average detection results of the left and right key points (E.g. left knee, right knee).(rows 353-358)

5. In my opinion, one of the most important challenges in recognizing a human pose is independence from the user. Therefore I think, that the results of leave-one-subject-out validation really characterize the robustness of the method. Please add such results to your paper

Our response: Our improved model is the same as most mainstream human pose estimation models which are independence from the user. The previous experiments on several datasets show that our improved model has very high detection accuracy. Besides, it could also achieve perfect results in more complex scenes. We add a new experiment that we trained our GoogLeNet13-CPM-Stage6 on the MPII Human Pose training dataset and then validated it on the MPII Human Pose validation dataset. Our improved model achieves a high level of accuracy compared with most other leading human pose estimation models on the MPII Human Pose validation dataset.(rows 382-383)

Reviewer 3 Report

The authors present an improved Convolutional Pose Machine for human pose estimation. Therefore, they change the feature extractor and use the more powerful GoogLeNet instead of the original feature extractor in order to obtain more accurate predictions with less parameters and thus faster computation time.

The structure of the paper is good, and mostly easy to follow. Also, the motivation behind the idea is clear. However, several terms and concepts are not properly defined and might require improvement.
The authors claim three innovations: (1) improved model with deeper network structure, (2) fine-tuning, and (3) the Inception structure. However, the second claim, ie. fine-tuning, is not elaborated in the paper and probably simply refers to using a network initialization pretrained on ImageNet for the changed network architecture, and the third claim, ie. the Inception structure, is essentially the same as the first claim, which is using GoogLeNet (=Inception v1) as network. The authors often claim that the new network has faster training time, but I see lack of motivation to reduce training time since during training one has essentially weeks of time as long as inference is efficient.

In terms of novelty, the only innovation I see is changing the feature extractor of the Convolutional Pose Machines [10]. With this respect, it would be interesting to evaluate more different feature extractors, such as ResNet [He et al. Deep Residual Learning for Image Recognition], Inception v3 [Szegedy et al. Rethinking the Inception Architecture for Computer Vision], etc., that might work even better than the proposed GoogLeNet.

Further more detailed remarks:
- L98: Please fix notation of "q-th".
- Sec. 2.1.1: The citations of [10] in this section are not correct and should differently refer to the paper of "Ramakrishna et al.: Pose Machines: Articulated Pose Estimation via Inference Machines"
- Fig. 1: The separation of the figure into (a) and (b) is not well visible. Probably use a vertical line to make that more clear.
- L130: What does "central constraints" mean?
- Fig. 2: What do the abbreviations C/P/MC denote? What is the "center map"? What are the losses? If the losses correspond to the ones two pages later, please denote so.
- Fig. 3: What does "depth 128" mean? Several heatmaps are shown.
- L151: What exactly is "VGG10-CPM-Stage6", which is referred to [10] but not described in the reference.
- L167: What does "Inc(3a)~Inc(4a)" denote?
- Fig. 4: The depicted architecture is not clear to me. What does "Different layers of Inception" mean? What are 4-3C, 4-4C, etc.? Please fix "lays".
- Table 1/2: What does "data change" mean? Do the authors refer to the network architecture in these tables?
- Fig. 5: What is the used dataset for this figure? What is the metric?
- Fig. 6: Please consider adjusting the scale of the y axis for better visibility.
- Fig. 5/6: It would be necessary to compare to the original CPM-Stage6 [10] as well.
- Table 3: The authors denote several parameters. But where are these parameters applied and used?
- L235: What is the unreferenced "step learning strategy"?
- Sec. 2.2.2: The authors "select the model with the best verification indicators". This seems like cherry-picking to me. What are the effects of this on the final results?
- L254: What is "reverse propagation"? Do the authors actually mean "backpropagation"?
- L312: The authors claim that the changed network has faster convergence, however, I could not find any plots of figures where this claim is supported. For example, an appropriate plot would show the validation error over the training iterations.
- Table 8: Is the evaluation of the runtime all done on the same hardware? What is "each" in the unit "ms/each"?
- Please consider improving the language and reading flow throughout the document.

Author Response

I am very grateful to receive your suggestions. These suggestions are very valuable and very helpful for the improvement of our manuscripts.We have revised the manuscript according to the your suggestion. Here are responses for the suggestion.

Suggestion 1. The authors claim three innovations: (1) improved model with deeper network structure, (2) fine-tuning, and (3) the Inception structure. However, the second claim, ie. fine-tuning, is not elaborated in the paper and probably simply refers to using a network initialization pretrained on ImageNet for the changed network architecture, and the third claim, ie. the Inception structure, is essentially the same as the first claim, which is using GoogLeNet (=Inception v1) as network. The authors often claim that the new network has faster training time, but I see lack of motivation to reduce training time since during training one has essentially weeks of time as long as inference is efficient.

Our response: We claim three innovations: (1) improved model with deeper network structure, (2) fine-tuning, and (3) the Inception structure. Actually, the (2) is a small innovations. The improved model applies fine-tuning strategy in our model training which benefits the estimation accuracy. As for (3), the(3) is belonging to (1). Because of the inaccuracy of my previous language expression, we have deleted the (3) now.(rows 82,84)

Suggestion 2. In terms of novelty, the only innovation I see is changing the feature extractor of the Convolutional Pose Machines [10]. With this respect, it would be interesting to evaluate more different feature extractors, such as ResNet [He et al. Deep Residual Learning for Image Recognition], Inception v3 [Szegedy et al. Rethinking the Inception Architecture for Computer Vision], etc., that might work even better than the proposed GoogLeNet.

Our response: In novelty, we also combine the CPMs with the Resnet to redesign some new models. Unfortunately, although the depth of the Resnet is deeper than GoogLeNet, the detection results of these new model are not ideal. At the same time, the parameters quantity of these new model is also larger. Besides, the Inception v2, Inception v3 and Inception v4 are also considered by us. Because the structure of GoogLeNet (Inception v1) is very different from them, we studied the structure of them carefully and find that it is not available to combine CPMs with them well .(rows 402-407).

For further more detailed remarks.

- L98: Please fix notation of "q-th".

Our response: The q is between 0 and 14 in this paper.

- Sec. 2.1.1: The citations of [10] in this section are not correct and should differently refer to the paper of "Ramakrishna et al.: Pose Machines: Articulated Pose Estimation via Inference Machines"

Our response: The citations of [10] in this section have been corrected now.(rows 108)

- Fig. 1: The separation of the figure into (a) and (b) is not well visible. Probably use a vertical line to make that more clear.

Our response: The figure (a) and (b) have been revised now. We use a vertical line to make that more clear.(rows 119))

- L130: What does "central constraints" mean?

Our response: It means that the center point of the human body picture, it is used to aggregate the response maps to the image centers.(rows 145-146)

- Fig. 2: What do the abbreviations C/P/MC denote? What is the "center map"? What are the losses? If the losses correspond to the ones two pages later, please denote so.

Our response: In the Figure 2, the “C”, ”MC1, MC2, ……” means different convolution layers, the “P” means different pooling layers. The “Center map” is the center point of the human body picture, it is used to aggregate the response maps to the image centers. The “Loss” is the minimum output cost function, it is the same as the ”Loss” in the Figure 4.(rows 132-136)(rows 176-178)

- Fig. 3: What does "depth 128" mean? Several heatmaps are shown.

Our response: The ”depth 128” means that the overlay of 128 heatmaps. There are 128 heatmaps. ( rows 147-148)

- L151: What exactly is "VGG10-CPM-Stage6", which is referred to [10] but not described in the reference.

Our response : There are two types of CPMs models designed by Shih-En Wei. One is the original CPM-Stage6 model described in the reference and the other is the VGG10-CPM-Stage6 model based on VGGNet-19 design exposed in author’s GitHub homepage. (They are both models that authors publicly expose).( rows 154-155)

- L167: What does "Inc(3a)~Inc(4a)" denote?

Our response: Because of the inaccuracy of my previous language expression, the “Inc(3a)~Inc(4a)” is replaced by ”Inc(4a)” (The first 9 layers of GoogLeNet). (rows 171-174. Specifically, we have selected different Inception layers, Inc(4a) (The first 9 layers of GoogLeNet), Inc(4b) (The first 11 layers of GoogLeNet), Inc(4c) (The first 13 layers of GoogLeNet), Inc(4d) (The first 15 layers of GoogLeNet), Inc(4e) (The first 17 layers of GoogLeNet), of GoogLeNet separately for stage 1 of new human pose estimation models.) (rows 226-227.)

- Fig. 4: The depicted architecture is not clear to me. What does "Different layers of Inception" mean? What are 4-3C, 4-4C, etc.? Please fix "lays".

Our response: The “Different layers of Inception" means “the Inc(4a), Inc(4b), Inc(4c), Inc(4d), Inc(4e) of the GoogLeNet”. Specifically, we have selected different Inception layers, Inc(4a) (The first 9 layers of GoogLeNet), Inc(4b) (The first 11 layers of GoogLeNet), Inc(4c) (The first 13 layers of GoogLeNet), Inc(4d) (The first 15 layers of GoogLeNet), Inc(4e) (The first 17 layers of GoogLeNet), of GoogLeNet separately for stage 1 of new human pose estimation models.(rows 171-174). In the Figure 4, the “C”, ”4-3C, 4-4C……”, ”MC1, MC2, ……” means different convolution layers, the “P” means different pooling layers. The “Center map” is the center point of the human body picture, it is used to aggregate the response maps to the image centers.(rows 176-178)

- Table 1/2: What does "data change" mean? Do the authors refer to the network architecture in these tables?

Our response: The data change means that the change of the feature maps with the depth of the network in Table1 and Table2 )

- Fig. 5: What is the used dataset for this figure? What is the metric?

Our response: The LSP dataset is used for this figure and we use [email protected] for evaluating the GoogLeNet13-CPM-Stage6. (rows 199-200)

- Fig. 6: Please consider adjusting the scale of the y axis for better visibility.

Our response :The Fig.6 is adjusted now for better visibility according to your suggestion. (rows 209)

- Fig. 5/6: It would be necessary to compare to the original CPM-Stage6 [10] as well.

Our response: We add the original CPM-Stage6 in Fig.5/6 to compare with other models now.(rows 203,209)

- Table 3: The authors denote several parameters. But where are these parameters applied and used?

Our response: These parameters in Table 3 are used for guiding the model training.(rows 251-252)

- L235: What is the unreferenced "step learning strategy"?

Our response: The “step learning strategy” is revised now. It means that the times of iterations required to adjust the learning rate.

- Sec. 2.2.2: The authors "select the model with the best verification indicators". This seems like cherry-picking to me. What are the effects of this on the final results?

Our response: It is part of the test of the experiment. We could get our best model by this.

- L254: What is "reverse propagation"? Do the authors actually mean "backpropagation"?

Our response: The “reverse propagation” actually means "backpropagation". We have revised it now.

- L312: The authors claim that the changed network has faster convergence, however, I could not find any plots of figures where this claim is supported. For example, an appropriate plot would show the validation error over the training iterations.

Our response:Because the verification or test requires separate code, so the accuracy of models detection is not verified during training. From the Table 6, we observed that our model has a faster convergence rate based on the time which it takes to complete the model training.(rows 338-339)

- Table 8: Is the evaluation of the runtime all done on the same hardware? What is "each" in the unit "ms/each"?

 Our response: the wei et al’s and our improved model* are evaluated on the same hardware environment, the speed of detection of other models is publicized by their authors. The “each” in the unit "ms/each" means ”each image”. The "ms/each" means the detection speed of each image.

Round 2

Reviewer 2 Report

All my comments and questions have been addressed. I believe the manuscript has been improved. I have no further questions.

Reviewer 3 Report

The authors improved the document and addressed my comments. I believe the manuscript can be accepted.